# Availability of evidence and comparative effectiveness for surgical versus drug interventions: an overview of systematic reviews and meta-analyses

Emmanuel A Zavalis [1,2] Anaïs Rameau,[3] Anirudh Saraswathula,[4] Joachim Vist,[1] Ewoud Schuit,[5,6] John P Ioannidis [2,7]

EAZ, AR and AS are joint first authors.

For numbered affiliations see end of article.

**Correspondence to**
Dr Emmanuel A Zavalis; emmanuel.zavalis@ki.se

## ABSTRACT

**Objectives** This study aims to examine the prevalence of comparisons of surgery to drug regimens, the strength of evidence of such comparisons and whether surgery or the drug intervention was favoured.

**Design** Systematic review of systematic reviews (umbrella review).

**Data sources** Cochrane Database of Systematic Reviews.

**Eligibility criteria** Systematic reviews attempt to compare surgical to drug interventions.

**Data extraction** We extracted whether the review found any randomised controlled trials (RCTs) for eligible comparisons. Individual trial results were extracted directly from the systematic review.

**Synthesis** The outcomes of each meta-analysis were resynthesised into random-effects meta-analyses. Egger's test and excess significance were assessed.

**Results** Overall, 188 systematic reviews intended to compare surgery versus drugs. Only 41 included data from at least one RCT (total, 165 RCTs) and covered a total of 103 different outcomes of various comparisons of surgery versus drugs. A GRADE assessment was performed by the Cochrane reviewers for 87 (83%) outcomes in the reviews, indicating the strength of evidence was high in 4 outcomes (4%), moderate in 22 (21%), low in 27 (26%) and very low in 33 (32%). Based on 95% CIs, the surgical intervention was favoured in 38/103 (37%), and the drugs were favoured in 13/103 (13%) outcomes. Of the outcomes with high GRADE rating, only one showed conclusive superiority in our reanalysis (sphincterotomy was better than medical therapy for anal fissure). Of the 22 outcomes with moderate GRADE rating, 6 (27%) were inconclusive, 14 (64%) were in favour of surgery and 2 (9%) were in favour of drugs. There was no evidence of excess significance.

**Conclusions** Though the relative merits of surgical versus drug interventions are important to know for many diseases, high strength randomised evidence is rare. More randomised trials comparing surgery to drug interventions are needed.

## INTRODUCTION

Many diseases are treated or managed with surgery. Some of them may also be addressed by pharmaceutical interventions and studying the effectiveness of these

## STRENGTHS AND LIMITATIONS OF THIS STUDY

⇒ The Cochrane database offers comprehensive coverage of health interventions with detailed methods sections that are likely to convey the intention to study surgical versus drug interventions even if no such randomised trials are found.

⇒ Journal-published systematic reviews outside of Cochrane were not considered, but these are unlikely to include topics where no eligible randomised trials are found.

⇒ We did not consider endovascular and endoscopic interventions in the surgery group and we did not consider non-pharmaceutical interventions in the control group.

⇒ We did not consider non-randomised observational studies, but these may have additional biases in estimating the outcomes of surgical versus drug interventions.

different interventions is important in optimising shared decision-making for patients and physicians. However, the amount and certainty of the evidence we hold in healthcare is limited,[1] and this situation is likely worse for surgical interventions due to serious challenges in running placebo-controlled or comparative effectiveness trials.[2] Challenges to controlled trials include unique patient anatomy, operator-dependent variables such as the skill or experience of the surgeon,[3–5] and the difficulty of successful blinding.[6] Due to these challenges, randomised controlled trials (RCTs) in surgery are less common than in non-surgical medical specialties. Although there have been calls to strengthen the quality of the evidence in surgery,[2 7 8] these have resulted in relatively few RCTs assessing surgical interventions, particularly in comparison to medical treatments.

A summary of the existing body, mapping the gaps of evidence on surgical versus medical interventions across diseases, does

not exist in the literature. A synthesis of this existing body of evidence is important to guide evidence-based care and inform decisions in the clinic where surgery and medical management are both reasonable options. We hypothesised that there may be a dearth of randomised evidence comparing surgery versus drugs and that even in topics where such RCTs exist, the evidence provided by them might be weak. To find RCTs comparing surgical versus pharmaceutical interventions, we conducted an umbrella review (an overview of systematic reviews)[9 10] by searching the Cochrane Database of Systematic Reviews for reviews considering comparisons of surgery to drugs. We aimed to examine the prevalence of intended comparisons of surgery to drug regimens, how often such comparisons had any RCTs, and, whenever RCTs were available, what was the strength of evidence of such comparisons, and whether surgery or the drug intervention was favoured.

## MATERIALS AND METHODS

This systematic review of systematic reviews (umbrella review) was structured based on the guidance provided by Belbasis *et al*[10] (for more information on reviews of reviews, see also Cochrane Handbook Chapter V: Overviews of Reviews[11]). For reporting, we adapted the Preferred Reporting Items for Systematic Reviews and Meta-Analyses guidelines[12] and the checklists are found as supplements. The protocol for the data collection and analysis was preregistered on the Open Science Framework website,[13] together with the raw data and code.

### Search strategy and selection criteria

We queried the Cochrane Database of Systematic Reviews using the term "surg*" in "Title/Abstract/Keywords" ("surg*(ti;ab;kw)") on 25 April 2022. Inclusion criteria for reviews were the search of RCTs comparing a surgical to a drug intervention.

A surgical intervention was defined as a procedural technique aiming to change anatomy to treat or alleviate a pathology or symptom (including dermatological excisions). We excluded endoscopic and endovascular procedures since many of them are performed by medical rather than surgical specialists. A drug intervention was defined as a treatment that used a non-supplement and non-vitamin, pharmaceutical agent. Dental procedures, radiation treatment and comparisons of surgery versus no treatment or only placebo were excluded from our study. Cochrane reviews that intended to compare surgical and pharmaceutical interventions were considered even in cases where the review was unsuccessful in finding any such comparisons.

As many surgical procedures also require drug regimens (eg, preoperatively or as background treatment), we allowed comparisons where the surgical arm including a drug intervention was compared with a drug intervention as well. Comparisons of surgery to surgery plus drugs were not eligible, as both arms used surgery.

The articles' abstracts were reviewed by EAZ and JV who coded the reviews independently for eligibility (include, exclude and unsure) first and then sought to reach a consensus among the reviews coded as unsure by either reviewer. If either reviewer included the review, it was included directly. The remaining differences were mediated by JPI, and a final check of all included studies was performed by JPI, EAZ and JV.

### Main outcomes

The main outcome assessed was the percentage of Cochrane systematic reviews that found eligible RCTs comparing head-to-head surgical and pharmacological interventions among all the reviews aiming to look for such studies. The strength of evidence of the existing comparison was also treated as a main outcome, as were the direction of effects in the review assessments, both in the original Cochrane analysis and our standardised reanalysis.

### Data extraction

EAZ extracted data for the included systematic reviews. The included systematic reviews were further classified into their corresponding surgical specialty field: cardiac surgery, dermatology, general surgery, neurosurgery, obstetrics and gynaecology, ophthalmology, orthopaedic surgery, otolaryngology, plastic surgery, thoracic surgery, urology and vascular surgery.

Whenever data were available from at least one RCT comparing a surgical to a drug arm, we identified the primary outcome(s) of the systematic review for the eligible comparison(s) by examining the methods section of the systematic review, and classified it as either mortality, composite or non-mortality. Data, in the form of contingency tables or means, SD and number of participants in each arms, from individual RCTs were then collected from Cochrane eligible reviews. We also collected available Grading of Recommendations, Assessment, Development, and Evaluations assessments (GRADE)[14] for the eligible comparisons and outcomes and the summary effect size as well as the 95% CI of the effect for the eligible comparison outcomes. Reviews that found no RCT of drugs to surgery were tabulated as having no data.

### Meta-analysis

As Cochrane reviewers may have used different statistical models in each topic to combine the results of RCTs in meta-analyses, we aimed for standardisation. To achieve it, we recalculated the summary effect size and heterogeneity for each topic using a random effects model following the Hartung-Knapp-Sidik-Jonkman approach[15 16] so that all outcomes/topics would be analysed with the same statistical methods. The modified Haldane-Anscombe continuity correction was used, that is, when studies had no event in either the surgical or the drug arm we added 0.5 to the entire contingency table of the specific study.[17]

The analysis of the data was performed using R V.4.1.3 (10 March 2022),[18] with the assessment of statistical

significance using a threshold for $\alpha$ of 0.005, as previously proposed.[19] The Wilson approach was used for CIs (99.5%) created for the primary outcomes.

### Additions to the protocol

The original preregistered protocol can be found at www. doi.org/10.17605/OSF.IO/3QVW9.

Some additions were made during the process of conducting this umbrella review. For each review, we noted the search date of the reviews to understand how old they may be. We assessed inter-rater reliability using Cohen's κ. We also probed for hints of bias by using the test of excess significance for each topic with two or more RCTs (and for the composite of observed and expected statistical significant results across all topics),[20] and small-study effects Egger's regression for meta-analyses with three or more RCTs.[21]

For each RCT in the included reviews, we extracted their year of publication to capture how recent the evidence was. Then, we extracted the specialty orientation of the journal, in which the RCT was published, using the categories 'mostly surgical', 'general' and 'mostly non-surgical'. The category 'mostly surgical' includes those journals that have 'surgery' in their title, those that have the name of a surgical specialty in their title and those affiliated with a surgical society. The category 'general' pertains to journals that cover all of medicine and its specialties, surgical and non-surgical. The category 'mostly non-surgical' includes all the remaining journals. We assessed whether the direction of effects (favouring surgery or favouring drug) was associated with the type of journal, hypothesising that RCTs published in mostly surgical journals may be more likely than other journals to favour surgery. We also examined whether the eligible RCTs that were included in the systematic reviews might have any overlap between different reviews. Finally, we extracted information on risk of bias assessments of the eligible RCTs, as these assessments had been performed in the Cochrane systematic reviews that had included the RCTs.

### Patient and public involvement

No patients were involved in the design and conduct of this umbrella review.

## RESULTS

### Search results

The selection flow chart for Cochrane systematic reviews is represented in figure 1. The search strategy retrieved 2495 articles from the Cochrane Database of Systematic Reviews. Among them, 440 were excluded by an automated search for withdrawn reviews and of studies with no mention of the word surgery and any of its variations in the abstract. Further manual assessment of titles and abstracts in duplicate resulted in 223 Cochrane reviews being potentially eligible. The inter-rater reliability was fair with a κ of 0.36 and 90% agreement on exclusion.

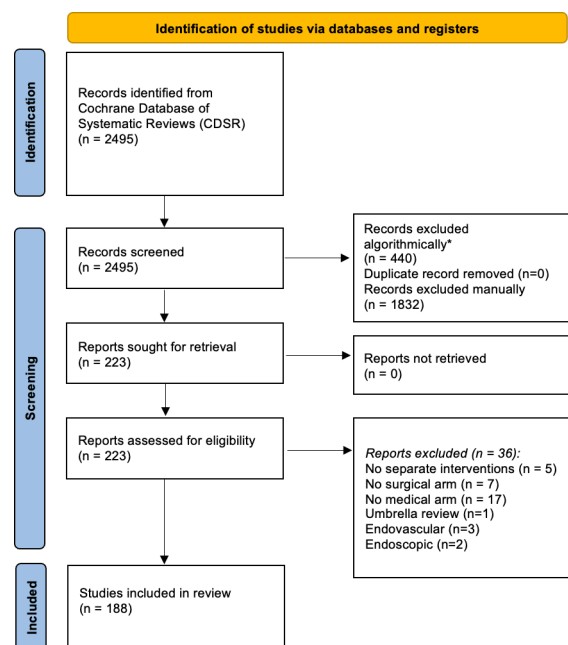

**Figure 1** PRISMA study selection flow chart. PRISMA, Preferred Reporting Items for Systematic Reviews and Meta-Analyses. *filtered for [surg*] in the abstract and removed withdrawn publications

All reviewer differences were in the articles classified as 'unsure' by either reviewer.

On full-text evaluation, 35 were excluded: in 5 reviews, the surgical and drug treatments were not in separate arms and hence they were not an eligible head-to-head comparison[22–26]; in 7 reviews, there was no surgical intervention arm[27–33]; in 17 reviews, there was no drug intervention[34–39 39–49]); 2 reviews were excluded for evaluating an endoscopic intervention[50 51]; 3 reviews were excluded for evaluating an endovascular intervention[52–54]; and finally, 1 review was excluded for being an umbrella review.[55]

Therefore, 188 Cochrane reviews were found to meet the inclusion criteria (online supplemental file 1). Of those, 147 Cochrane reviews aimed to investigate surgical versus drug interventions but were unable to find any RCTs meeting their selection criteria. The remaining 41 reviews contained data for at least one RCT in at least one head-to-head comparison of a surgical versus a drug intervention arm (22% (99.5% CI 14% to 31%)).

The 188 reviews covered all major surgical specialties (online supplemental table 1), with the most commonly represented specialties being general surgery (n=35), obstetrics and gynaecology (n=31), ophthalmology (n=25), orthopaedic surgery (n=23) and otolaryngology (n=23). When examining whether any specialty had compared surgery to drugs more than others, no significant difference was found (Fisher's exact p=0.62).

### Eligible RCTs for surgery versus drug comparisons

The 41 eligible reviews with data included 103 comparisons of surgery versus drug treatments with data on various primary outcomes (table 1), and they included data from a total of 165 RCTs with a total of 295 primary outcome

**Table 1** Eligible comparisons of surgical versus medical interventions

| Surgical arm | Drug arm | Disease | No of outcomes (studies) |
|---|---|---|---|
| Cardiac surgery | | | |
| Transmyocardial lazer revascularisation | Continued medication | Refractory angina | 3 (7,7,6) |
| Surgical closure | IV indomethacin | Patent ductus arteriosus | 1 (1) |
| Dermatology | | | |
| Surgical excision | Imiquimod | BCC | 4 (1,1,1,1) |
| Surgical excision | MAL-PDT | BCC | 3 (1,2,2) |
| Surgical excision | ALA-PDT | BCC | 2 (1,1) |
| General surgery | | | |
| Lateral internal sphincterotomy | Medical therapy (mainly GTN Isosorbide dinitrate and Botox) | Anal fissure | 1 (15) |
| Pancreatic resection | Chemoradiotherapy | Pancreatic cancer | 1 (2) |
| Oesophagectomy | Chemoradiotherapy and/or radiotherapy | Oesophageal cancer | 5 (5,3,1,1,1) |
| Laparoscopic fundoplication | Protein pump inhibitors | GERD | 5 (3,3,4,3,2) |
| Surgery | Tamoxifen | Primary breast cancer | 1 (3) |
| Neurosurgery | | | |
| Decompressive surgery | Prednisolone | Leprosy | 4 (1,1,1,1) |
| Epilepsy surgery | Continued antiepileptic drugs | Epilepsy | 2 (2,1) |
| Decompressive craniectomy | Medical treatment (including barbiturates) | High ICP in closed TBI | 2 (3,3) |
| Surgical decompression | Osmotic agents, blood pressure control and glucose control | Cerebral oedema in acute ischaemic stroke | 1 (3) |
| Surgical decompression | Dexamethasone, antihypertensives and intermittent diuresis | Primary supratentorial intracerebral haemorrhage | 1 (9) |
| Obstetrics and gynaecology | | | |
| Suction aspiration | Vaginal suppositories or im inj. of 9-methylene-PGE2 | Abortion | 3 (2,2,1) |
| Suction aspiration | Misoprostol | Abortion | 2 (22,9) |
| Suction aspiration | Vaginal or oral misoprostol | Abortion | 3 (15,13,5) |
| Suction aspiration | Misoprostol and mifepristone | Abortion | 2 (2,1) |
| Dilatation and curretage | Misoprostol | Abortion | 2 (1,2) |
| Dilation and evacuation | Misoprostol | Abortion | 1 (1,1) |
| Laparoscopic ovarian drilling | Medical ovulation induction | Infertility due to PCOS | 2 (9,14) |
| Laparoscopic ovarian drilling | Letrozele | Infertility due to PCOS | 2 (3,1) |
| Laparoscopic ovarian drilling | Gonadotropins | PCOS | 2 (1,1) |
| Laparoscopic ovarian drilling | Metformin, clomiphene | PCOS | 1 (2) |
| Laparoscopic ovarian drilling | Letrozele | PCOS | 1 (1) |
| Laparoscopic ovarian drilling | Metformin, letrozele | PCOS | 1 (1) |
| Laparoscopic ovarian drilling | Metformin | PCOS | 2 (2,1) |
| Transcervical resection of endometrium using rollerball coagulation | Hormone therapy or antifibrinolytic | Heavy menstrual bleeding | 7 (1,1,1,1,1,1,1) |
| Ophthalmology | | | |
| Amniotic membrane transplantation and medication | Lubrication, antibiotics and pressure lowering medication | Acute ocular burns | 1 (1) |
| Laser surgery | Intravitreal anti-VEGF | Pathological myopia | 2 (1,1) |

Continued

**Table 1** Continued

| Surgical arm | Drug arm | Disease | No of outcomes (studies) |
|---|---|---|---|
| iStent | Latanoprost/timolol | Open angle glaucoma | 1 (2) |
| Argon laser trabeculoplasty | IOP reducing medication | Open angle glaucoma | 3 (3,2,2) |
| Surgical correction | Botulinum toxin | Strabismus | 2 (2,1) |
| Orthopaedic surgery | | | |
| Open section of the carpal ligament | NSAID and splinting or corticosteroid injections | Carpal tunnel syndrome | 1 (2) |
| Open surgery | Corticosteroid injection | Trigger finger | 1 (2) |
| Decompressive surgery with or without fusion | Epidural steroid injection | Lumbar spinal stenosis | 3 (1,1,1) |
| Open unilateral sympathectomy (L2–4) | Intravenous prostanoid iloprost | Critical limb ischaemia | 1 (1) |
| Surgical rotator cuff repair | Non-operative treatment including corticosteroid injection and exercise | Rotator cuff tear | 1 (1) |
| Arthroscopic surgery | Sclerosing injection | Jumper's knee | 3 (1,1,1) |
| Otolaryngology | | | |
| Surgical orbital decompression | Intravenous methylprednisolone 1×3 followed by oral prednisolone | Thyroid eye disease | 1 (1) |
| Grommets (ventilation tubes) | Antibiotic prophylaxis | Recurrent acute otitis media | 1 (2) |
| Tonsillectomy or adrenotonsillectomy | Watchful waiting with or without analgesics and antibiotics | Tonsillitis | 5 (5,4,5,2,2) |
| Thoracic surgery | | | |
| Open thoracotomy | Thoracostomy drainage (with fibrinolytics) | Pleural empyema | 1 (1) |
| VATS | Thoracostomy drainage (with fibrinolytics) | Pleural empyema | 1 (7) |
| Urology | | | |
| Surgical reimplantation of ureters | Antibiotics | Primary vesicoureteric reflux | 1 (1) |
| Vascular surgery | | | |
| Carotid endarterectomy and aspirin 325 mg daily | Aspirin 325 mg daily | Asymptomatic carotid stenosis | 1 (2) |
| Aspirin and carotid surgery | Aspirin | Carotid stenosis | 2 (3,3) |
| Saphenofemoral disconnection | Therapeutic LMWH | Superficial thrombophlebitis | 2 (1,1) |
| Surgery including primary amputation | Thrombolysis (w/rt-Pa or urokinase) | Acute limb ischaemia | 1 (3) |

BCC, basal cell carcinoma of the skin; GERD, gastro-oesophageal reflux disease; GTN, glyceryl tri-nitrate; ICP, intra-cranial pressure; IOP, intraocular pressure; LMWH, low molecular weight heparin; MAL-PDT, Methyl aminolevulinate photodynamic therapy; NSAID, nonsteroidal anti-inflammatory drugs; PCOS, polycystic ovarian syndrome; TBI, traumatic brain injury; VEGF, vascular endothelial growth factor.

assessments. For the 165 trials, the median publication year was 2005 and the IQR was 1994–2016. The median search date year of the eligible reviews was 2016 (IQR 2010–2022). 19 of the 165 trials were part of two different Cochrane reviews. 14 of these 19 trials also overlapped in terms of addressing the same outcome and treatment arms. The overlapping studies comprised >50% of the included RCTs in 2 of 103 meta-analyses.

### Risk of bias in eligible RCTs

Risk of bias assessments of the 165 eligible RCTs by the authors of the original Cochrane systematic reviews did not always include the same elements. Specifically, for the generation of the randomisation sequence, information had been extracted in 141 trials and of those 6 (4%) were deemed to be at high risk of bias, 42 (30%) were unclear and 93 (66%) were at low risk of bias. The respective numbers were 9 (6%) high risk, 63 (39%) unclear and 89 (55%) low risk among 161 RCTs extracted for risk of allocation bias; 101 (73%) high risk, 29 (21%) unclear and 9 (6%) low risk among 139 RCTs extracted for performance bias; 47 (34%) high risk, 71 (51%) unclear and 21 (15%) low risk among 139 RCTs extracted for detection

**Table 2** Comparisons where the surgical treatment was superior to the drug treatment

| Surgical arm | Drug arm | Disease | Outcome | Treatment effect (95% CI) | GRADE assessment |
|---|---|---|---|---|---|
| Transmyocardial lazer revascularisation | Continued medication | Refractory angina | Angina reduction | OR=4.63 (3.43 to 6.25) | Low |
| Surgical excision | Imiquimod | BCC | Recurrence (3 years) | RR=0.1 (0.03 to 0.31) | Moderate |
| | | | Recurrence (5 years) | RR=0.13 (0.05 to 0.36) | Moderate |
| Surgical excision | MAL-PDT | BCC | Recurrence (3 years) | RR=0.04 (0 to 0.61) | Low |
| Surgical excision | ALA-PDT | BCC | Recurrence (3 years) | RR=0.09 (0.02 to 0.38) | Moderate |
| | | | Recurrence (5 years) | RR=0.08 (0.02 to 0.34) | Moderate |
| Laparoscopic fundoplication | Protein pump inhibitors | GERD | GORD-specific QOL (<1 years) | SMD=0.58 (0.46 to 0.7) | Low |
| Lateral internal sphincterotomy | Medical therapy (mainly GTN and Botox) | Anal fissure | Non-healing (persistence or recurrence) 2 months. | OR=0.11 (0.06 to 0.23) | High |
| Epilepsy surgery | Continued antiepileptic drugs | Epilepsy | Proportion (%) free from seizures (1 year) | RR=9.78 (4.73 to 20.2)* | Low |
| | | | Proportion free from all seizures including auras (1 year) | RR=15 (2.08 to 108.23) | Very low |
| Surgical decompression | Osmotic agents, blood pressure control and glucose control | Cerebral oedema in acute ischaemic stroke | Death at the end of follow-up | OR=0.19 (0.09 to 0.37) | |
| Surgical decompression | Dexamethasone, antihypertensives and intermittent diuresis | Primary supratentorial intracerebral haemorrhage | Death or dependence at end of follow-up | OR=0.71 (0.58 to 0.88) | |
| Suction aspiration | Misoprostol | Abortion | Complete miscarriage | RR=1.11 (1.06 to 1.17) | Very low |
| | | | Complete miscarriage | RR=1.04 (1.02 to 1.06) | Very low |
| Dilatation and curettage | Misoprostol | Abortion | Complete miscarriage | RR=1.18 (1.1 to 1.27)* | Very low |
| Dilatation and evacuation | Misoprostol | Abortion | Combined major and minor complications | OR=0.12 (0.03 to 0.46) | |
| Laparoscopic ovarian drilling | Medical ovulation induction | Infertility due to PCOS | Multiple pregnancy | OR=0.34 (0.18 to 0.66) | Moderate |
| Laparoscopic ovarian drilling | Gonadotropins | PCOS | Menstrual regularity at 6 months | OR=19.2 (3.17 to 116) | Very low |
| Transcervical resection of endometrium using rollerball coagulation | Hormone therapy or antifibrinolytic | Heavy menstrual bleeding | Control of bleeding (cure or improvement to acceptable level) 4 months. | RR=2.66 (1.94 to 3.64) | Moderate |
| | | | Control of bleeding (cure or improvement to acceptable level) 2 years | RR=1.29 (1.06 to 1.57) | Low |
| | | | Overall satisfaction with treatment 4 months. | RR=2.8 (1.96 to 3.99) | Moderate |
| | | | Overall satisfaction with treatment 2 years | RR=1.4 (1.13 to 1.74) | Moderate |
| | | | Adverse events at 4 months | RR=0.26 (0.15 to 0.46) | Moderate |
| Surgical correction | Botulinum toxin | Strabismus | Improved ocular alignment >10 dioptres, adults | RR=2.63 (1.18 to 5.9) | Low |
| iStent | Latanoprost/timolol | Open angle glaucoma | Proportion of participants who were drop-free 6–18 months | RR=125 (17.8 to 884) | Very low |

Continued

**Table 2** Continued

| Surgical arm | Drug arm | Disease | Outcome | Treatment effect (95% CI) | GRADE assessment |
|---|---|---|---|---|---|
| Argon laser trabeculoplasty | IOP reducing medication | Open angle glaucoma | Failure to control IOP | RR=0.8 (0.71 to 0.91) | |
| Arthroscopic surgery | Sclerosing injection | Jumper's knee | Knee pain (0–100, 12 months) | MD=−28.3 (−41.79 to −14.81) | Low |
| | | | Participant global assessment of success (1–100, 12 months) | MD=33.9 (18.74 to 49.06) | Low |
| Decompressive surgery with or without fusion | Epidural steroid injection | Lumbar spinal stenosis | Zurich claudication questionnaire (symptom evaluation) 6 weeks | MD=−0.6 (−0.77 to −0.43) | Low |
| Open unilateral sympathectomy (L2-4) | IV prostanoid iloprost | | Complete ulcer healing w/o rest pain or major amputation (24 weeks) | RR=1.76 (1.35 to 2.29) | Low |
| Grommets (ventilation tubes) | Antibiotic prophylaxis | Recurrent acute otitis media | Proportion of patients who have no recurrences (6 months) | RR=1.68 (1.07 to 2.65)* | Very Low |
| Tonsillectomy or adrenotonsillectomy | Watchful waiting with or without analgesics and antibiotics | Tonsillitis | Episodes of sore throat of any severity (children) | MD=−0.56 (−1.04 to −0.07)* | Moderate |
| | | | Sore throat days (children) | MD=−5.13 (−8.03 to −2.2)* | Moderate |
| | | | Episodes of sore throat of any severity (adults) | MD=3.61 (−7.92 to −0.7)* | Moderate |
| | | | Sore throat days (adults) | MD=−10.64 (−15.52 to −5.76)* | Moderate |
| Aspirin and carotid surgery | Aspirin | Carotid stenosis | Any stroke or operative death | RR=0.85 (0.77 to 0.95)* | Moderate |

*Our reanalysis using a random effects meta-analysis model shows that the 95% CI includes the null (results are inconclusive).
BCC, basal cell carcinoma of the skin; GERD, gastro-oesophageal reflux disease; GNT, glyceryl trinitrate; GRADE, grading of recommendations, assessment, development, and evaluations; IOP, intraocular pressure; MD, mean difference; PCOS, polycystic ovarian syndrome; QOL, quality of life; RR, risk ratio; SMD, standardised mean difference.

bias; 20 (16%) high risk, 15 (12%) unclear and 90 (72%) low risk among 125 RCTs extracted for attrition bias; 17 (12%) high risk, 56 (41%) unclear and 64 (47%) low risk among 137 RCTs extracted for reporting bias, and 17 (13%) high risk, 29 (23%) unclear and 80 (64%) low risk among 126 extracted for other risk of bias.

### Comparative effectiveness of surgery versus drugs

Based on the 95% CI of the summary estimate obtained by the Cochrane review authors, surgery was more effective in 36 of the 103 outcomes of various comparisons (35% (99.5% CI 23% to 49%)), and drugs were more effective in 15 (15% (99.5% CI 6% to 26%)). Fifty-two (50% (99.5 CI% 37% to 64%)) outcomes were inconclusive. The respective numbers were 1/12 (8%), 1/12 (8%) and 10/12 (83%) for mortality outcomes; 3/11 (27%), 3/11 (27%) and 5/11 (46%) for composite outcomes; and 32/80 (40%), 11/80 (14%) and 37/80 (46%) for non-mortality outcomes.

When we standardised the meta-analyses to use the same random effects method for all analyses, surgery was favoured in 28/103 outcomes (32%), drugs were favoured in 9/103 (10%) outcomes and 66/103 (58%) outcomes were inconclusive. The respective numbers

were 1/12 (8%), 0/12 (0%) and 11/12 (92%) for mortality outcomes; 3/11 (18%), 2/11 (27%) and 6/11 (55%) for composite outcomes and 24/80 (30%) 7/80 (9%) and 49/80 (61%) for non-mortality outcomes.

Table 2 shows the topics for which the surgical intervention was found to be more effective and table 3 shows those where the drug arm was found to be more effective, all according to the Cochrane authors' analysis. Online supplemental table 2 does the same for the topics for which the comparisons were inconclusive.

### Tests of bias and heterogeneity

Of the 103 comparisons, only 31 had ≥3 studies to be able to run an Egger regression for small study effects and only 5 had at least 10 studies to allow a meaningful application of this regression test. 3/5 with 10 or more studies had a small study effects signal suggestive of potential publication bias (p<0.05); all 3 compared surgical to pharmacological methods of abortion. The test of excess significance applied to all outcomes with ≥2 studies gave signals of potential bias in 16/53 outcomes (245 individual study outcomes) and across all outcomes the expected number of statistically significant results was 74 vs an observed 84 across 245 study outcomes (p=0.27).

**Table 3** Comparisons where the drug treatment was superior to the surgical treatment

| Surgical arm | Drug arm | Disease | Outcome | Treatment effect (95% CI) | GRADE assessment |
|---|---|---|---|---|---|
| Surgical excision | Imiquimod | BCC | Observer-rated good/excellent cosmetic outcome | RR=0.59 (0.47 to 0.74) | Low |
| Surgical excision | MAL-PDT | BCC | Observer-rated good/excellent cosmetic outcome | RR=0.85 (0.79 to 0.92)* | Moderate |
| Surgical excision | MAL-PDT | BCC | Patient-rated good/excellent cosmetic outcome | RR=0.53 (0.44 to 0.65)* | Moderate |
| Oesophagectomy | Chemoradiotherapy and/or radiotherapy | Oesophageal cancer | Serious adverse event (3 months) | RR=1.73 (1.11 to 2.67)* | Very low |
| | | | Short-term health-related QOL | MD=0.93 (0.24 to 1.62) | Very low |
| Laparoscopic fundoplication | Protein pump inhibitors | GERD | Serious adverse events | RR=1.46 (1.01 to 2.11) | Very low |
| Pancreatic resection | Chemoradiotherapy | Pancreatic cancer | Overall mortality (5 years) | HR=2.63 (1.72 to 4)* | Very low |
| Laparoscopic ovarian drilling | Medical ovulation induction | Infertility due to PCOS | Live birth | OR=0.71 (0.54 to 0.92) | Low |
| Suction aspiration | Vaginal or oral misoprostol | Abortion | Surgical evacuation | RR=20 (9.1 to 50) | Very low |
| Laser surgery | Intravitreal anti-VEGF | Pathological myopia | Change in best-corrected visual acuity | MD=0.22 (0.01 to 0.43)* | Low |
| Amniotic membrane transplantation and medication | Lubrication, Antibiotics and Pressure lowering medication | Acute ocular burns | Visual acuity at final follow-up | MD=−0.83 (-1.32 to −0.34) | Very low |
| Decompressive surgery with or without fusion | Epidural steroid injection | Lumbar spinal stenosis | Oswestry Disability Index 6 weeks | MD=5.7 (0.57 to 10.83) | Low |
| | | | Pain intensity (VAS) 6 weeks | MD=2.4 (1.92 to 2.88) | Low |
| Tonsillectomy or adrenotonsillectomy | Watchful waiting with or without analgesics and antibiotics | Tonsillitis | Episodes of moderately or severely sore throat (children) | MD=0.62 (0.22 to 1.03)* | Low |
| Carotid endarterectomy and Aspirin 325 mg daily | Aspirin 325 mg daily | Asymptomatic carotid stenosis | Perioperative stroke or death, or stroke of any territory or type during follow-up | RR=6.49 (2.53 to 16.61) | |

*Our reanalysis using a random effects meta-analysis model shows that the 95% CI includes the null (results are inconclusive).
BCC, basal cell carcinoma of the skin; GERD, Gastro-oesophageal reflux disease; GRADE, grading of recommendations, assessment, development, and evaluations; MD, mean difference; PCOS, polycystic ovarian syndrome; QOL, quality of life; RR, risk ratio; VAS, visual analogue scale.

Among the 50 topics with 2 or more studies, the median of $I^2$ was 43% (IQR 0%–80%).

## Strength of evidence according to GRADE

GRADE assessment of the strength of the evidence showed high rating for 4 outcomes (4%), moderate for 22 (21%), low for 27 (26%) and very low for 33 (32%). No GRADE assessment was performed for 17 (17%) outcomes.

According to GRADE assessments, only cardiac surgery, obstetrics and gynaecology and general surgery interventions had high GRADE ratings. Otolaryngology and dermatology had many moderate ratings. Almost all other GRADE ratings were low or very low (table 4).

Of the four outcomes with high GRADE rating, sphincterotomy for anal fissure showed superiority over medical treatment while the other three comparisons were inconclusive. Of the 22 outcomes with moderate GRADE rating, 6 (27%) were inconclusive, 14 (64%) were in favour of surgery and 2 (9%) were in favour of the drug regimen according to the calculations of the Cochrane authors (14 (64%), were inconclusive, 7 (32%) favoured the surgical arm and 1 (5%) were in favour of the drug regimen according to our standard random-effects calculations).

## Results of RCTs according to journal of publication

Of the 165 eligible RCTs (295 outcome assessments), 73 RCTs (133 assessments) were published in mostly surgical journals, 38 RCTs (69 assessments) in general journals and 54 RCTs (93 assessments) in mostly non-surgical journals. Based on 95% CIs for the assessments of RCTs

**Table 4** GRADE assessment across specialties

| Specialty | Very low | Low | Moderate | High | None available |
|---|---|---|---|---|---|
| Cardiac surgery | 0 (0) | 1 (25) | 0 (0) | 2 (50) | 1 (25) |
| Dermatology | 0 (0) | 3 (33) | 6 (67) | 0 (0) | 0 (0) |
| General surgery | 9 (69) | 3 (23) | 0 (0) | 1 (8) | 0 (0) |
| Neurosurgery | 5 (50) | 2 (20) | 1 (10) | 0 (0) | 2 (20) |
| Obstetrics and gynaecology | 14 (45) | 4 (13) | 7 (23) | 1 (3) | 5 (16) |
| Ophthalmology | 2 (20) | 5 (50) | 0 (0) | 0 (0) | 3 (30) |
| Orthopaedic surgery | 2 (20) | 6 (60) | 1 (10) | 0 (0) | 1 (10) |
| Otolaryngology | 1 (14) | 1 (14) | 4 (57) | 0 (0) | 1 (14) |
| Thoracic surgery | 0 (0) | 1 (50) | 1 (50) | 0 (0) | 0 (0) |
| Urology | 0 (0) | 0 (0) | 0 (0) | 0 (0) | 1 (100) |
| Vascular surgery | 0 (0) | 1 (17) | 2 (33) | 0 (0) | 3 (50) |

GRADE, grading of recommendations, assessment, development, and evaluations.

published in mostly surgical journals, 40/133 (30%) were in favour of surgery, 14/133 (11%) were in favour of drugs and 79/133 (59%) were inconclusive. The respective numbers for the assessments of RCTs published in general journals were 27/69 (39%), 5/69 (7%) and 37/69 (53%); and for the assessments of RCTs published in mostly non-surgical journals they were 22/93 (24%), 15/93 (16%) and 56 (60%), respectively. The proportion of RCTs favouring surgery was not significantly higher in mostly surgical journals (30%) compared with other journals (39% and 24% for general and non-surgical journals, respectively) (p=0.18 by Fisher's exact test).

## DISCUSSION
### Main findings
In a subset of Cochrane reviews that aimed to compare surgery to drugs we found that only one in five systematic reviews that had shown interest in such comparisons eventually found data from any RCTs for comparisons of the two modes of interventions. Furthermore, the majority of the comparisons where RCTs of surgery versus drugs had inconclusive results, few studies per meta-analytical outcome (30% with 3 or more studies) and also had low or very low strength of the evidence on GRADE assessments, and many trials had high risk of performance and detection bias.

Anal fissure was the only disease in our sample that had high GRADE evidence and a direction of effect indicating that one intervention (sphincterotomy) was more effective. Consequently, in the vast majority of cases where surgical and pharmaceutical interventions are available for treatment, an evidence-based decision in the clinic is difficult. Our secondary post hoc analysis of the type of journal where the eligible RCTs were published showed that results published in surgical journals were not necessarily more prone to favour the surgical arm of an RCT over the pharmaceutical arm.

### Strengths
This study covers the entire Cochrane database which is considered a high-quality comprehensive collection of systematic reviews. Cochrane reviews tend to address questions typically asked in routine clinical practice and underpin many clinical guideline recommendations, making this sample all the more relevant to everyday practice.[56] Another strength of this study is that all surgical specialties were included. This is, therefore, to our knowledge the first project aiming to assess the extent of comparative evidence for surgery versus pharmacotherapy for a diverse spectrum of diseases.

### Limitations
Our analysis has several limitations. First, our predefined inclusion criteria excluded non-pharmacological medical interventions. Several comparisons may be found in the literature where surgery is compared against non-surgical non-pharmacological medical interventions, such as with continuous positive airway pressure (CPAP) or radiotherapy. We also excluded endovascular and endoscopic procedures since they may be performed by surgical and medical specialists. These eligibility choices aimed to achieve some homogeneity in a project that is by definition already very heterogeneous. The use of an algorithm to filter out papers with no mention of the word surgery as well as the search strategy itself may have led to us missing reviews that discuss a particular surgical procedure but never explicitly mention the word surgery but merely the name of the intervention.

Second, we focused exclusively on RCTs, but other types of evidence, for example, non-RCTs, or uncontrolled clinical trials may also exist and sometimes their results may be compelling enough to deem a randomised study unnecessary. Such unquestionable superiority in the absence of randomised evidence is however unlikely.[57] Efforts such as IDEAL[8] have laid out much of the groundwork

for performing RCTs in surgical research, yet a dearth of RCTs in the surgical realm of research persists to this day.

Third, only one database (Cochrane Database of Systematic Reviews) was used for this study, and we did not examine non-Cochrane meta-analyses published as journal articles. While the database aims to be all inclusive, there are still some topics in medical and surgical care that have not been covered by Cochrane reviews.

However, the Cochrane database is more meticulous in describing its methods and it will routinely publish systematic reviews that have found no eligible articles, while this is unlikely in systematic reviews published in traditional journals. Therefore, including systematic reviews from journals may have distorted the picture and also caused a problem of overlapping systematic reviews. Moreover, we did not assess the methodological rigour or reporting quality of the Cochrane systematic reviews,[58] as this was not the focus of our study. Cochrane systematic reviews score very highly in standard tools like the assessing the methodological quality of systematic reviews tool (AMSTAR),[59] both because they are very meticulous and also because AMSTAR and AMSTAR-2 were developed with inspiration from the Cochrane Handbook.

Fourth, it is possible that within the same disease, subgroups of patients may be eligible only for medical or only for surgical treatment, or that one or the other approach is much better only for specific subgroups. With the dearth of evidence we found for the overall analysis, identification of such subgroup effects would be unlikely and error-prone.

### Context of these findings

Sequestration between different disciplines and specialties[60] may lead to isolation of specialists who use different tools, and this may lead to a lack of comparisons of the treatments that each specialty uses. Each specialty may have its own community, journals, meetings and research agenda, limiting communication between different specialists even though they may be dealing with the same disease from different angles and with different therapeutic sets. This lack of communication may also be due to differences in mentorship and the trend of subspecialisation in medical training separating clinicians and their practices even further,[61] or to differing incentive structures.

Prior literature comparing surgical and medical interventions has assessed specific treatments, such as that for basal cell carcinoma,[60] and demonstrated that sequestration was prominent. Despite a large number of trials, almost all of them compared medical interventions among themselves, or surgical interventions among themselves, rather than comparing between these two groups of treatment even though both groups of treatment could have been used. Our work shows that this issue of sequestration is widespread in surgical versus pharmaceutical interventions, and that even where comparisons exist, there are too few, as well as often biased trials.

### CONCLUSION

This study suggests that comparisons of pharmaceutical and surgical interventions are infrequent. The available comparisons have very few included studies which makes heterogeneity, and bias hard to quantify and may yield spurious results with the normality assumptions underpinning common frequentist meta-analytical approaches.[62] That is, even for the comparisons that have been retrieved the evidence is not sufficient.

Even accepting the difficulties in performing RCTs involving surgical interventions, our results still indicate a need for more comparative effectiveness research and for improved communication between surgical and medical specialties to bridge this gap in evidence. There are, of course, barriers to this. Head-to-head comparisons of treatments are often disfavoured by manufacturers leery of jeopardising their product against that of a competitor,[63 64] and incentives unfortunately exist for both surgical and medical practitioners to promote treatments they are able to offer. Moving forward, both medical and surgical professional societies should collaborate to design fair and unbiased trials, and funders should also keep such research on their radars to try and overcome these structural obstacles.

### Future research

Future clinical research should try to expand the scope, volume and methodological rigour of comparative evidence on surgical versus medical interventions. This work should involve both surgical and medical specialists and should also incorporate patient preferences. Long-term patient-centred outcomes, including both benefits and harms, should become available to put surgical and medical practices into proper perspective.

**Author affiliations**
[1]Department of Learning Informatics Management and Ethics, Karolinska Institutet, Stockholm, Sweden
[2]Meta-Research Innovation Center at Stanford (METRICS), Stanford University, Stanford, California, USA
[3]Sean Parker Institute for the Voice, Department of Otolaryngology–Head and Neck Surgery, Weill Cornell Medical College, New York, New York, USA
[4]Department of Otolaryngology–Head and Neck Surgery, The Johns Hopkins University School of Medicine, Baltimore, Maryland, USA
[5]Julius Center for Health Sciences and Primary Care, University Medical Center Utrecht, Utrecht University, Utrecht, The Netherlands
[6]Cochrane Netherlands, University Medical Center Utrecht, Utrecht University, Utrecht, The Netherlands
[7]Stanford Prevention Research Center, Department of Medicine, and Department of Epidemiology and Population Health, Stanford University, Stanford, California, USA

**Correction notice** This article has been corrected since it was published. There was an error in the affiliation of Ewoud Schuit, which has now been corrected.

**Contributors** AR, AS, EAZ and JPAI developed the idea, EAZ and JPAI interpreted the review data, JV and EAZ extracted the data. ES aided in the statistical analysis. All authors reviewed the manuscript and have edited and approved the submission. JPAI is the guarantor of this work.

**Funding** The work of John Ioannidis has been funded by an unrestricted gift from Sue and Bob O'Donnell. AR is supported by a Paul B. Beeson Emerging Leaders Career Development Award in Aging (K76 AG079040) from the National Institute on Aging and by the Bridge2AI award (OT2 OD032720) from the NIH Common Fund. AS was supported by the National Institute on Deafness and Other Communication

Disorders training grant 2T32DC000027. ES gratefully acknowledges financial contribution for his research by the Netherlands Organisation for Scientific Research (project 825.14.001).

**Competing interests** AR is a medical advisor for Perceptron Health.

**Patient and public involvement** Patients and/or the public were not involved in the design, or conduct, or reporting, or dissemination plans of this research.

**Patient consent for publication** Not applicable.

**Provenance and peer review** Not commissioned; externally peer reviewed.

**Data availability statement** Data are available in a public, open access repository. The dataset supporting the conclusions of this article, and the used code is available in the Open Science Framework repository https://www.doi.org/10.17605/OSF.IO/RK7HU.

**ORCID iDs**
Emmanuel A Zavalis http://orcid.org/0000-0001-6205-1362
John P Ioannidis http://orcid.org/0000-0003-3118-6859

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
