## [Reviewer comments · BMJ Open]

ARTICLE DETAILS

TITLE (PROVISIONAL)	Availability of evidence and comparative effectiveness for surgical versus drug interventions: an overview of systematic reviews and meta-analyses
AUTHORS	Zavalis, Emmanuel A.; Rameau, Anaïs; Saraswathula, Anirudh; Vist, Joachim; Schuit, Ewoud; Ioannidis, John

VERSION 1 – REVIEW

REVIEWER	Bajaj, Jitin NSCBM Medical College
REVIEW RETURNED	10-Jul-2023

GENERAL COMMENTS	What was the main aim and hypothesis of the study? In the table 2 showing comparison between the medical and surgical arm for epilepsy surgery, why you have shown a low grade assessment even when three randomized controlled trials are already there showing surgery superior to the medical arm in drug-resistant epilepsy. How did you defined the grade assessment? Can you give references in the results of the respective RCTs which you found? What benefit to the literature you seem to seek with your findings?
--

REVIEWER	Butera, Gisela National Institutes of Health, National Institutes of Health Library
REVIEW RETURNED	18-Aug-2023

GENERAL COMMENTS	Overall comments: Overall, the review highlights the need to assess the level of evidence available on surgical versus medical treatments. The authors successfully described why they conducted an overview of reviews to identify the evidence but there are areas especially within the methodology and reporting of results that needs improvement. Introduction section: Page 7: Authors nicely identified the gaps in the literature and the challenges in the quality of the evidence. Page 7/Lines 44-51: There are concerns with searching only one database. If the aim of your review is to identify RCTs comparing surgical vs. pharmaceutical interventions, you should include additional databases. In addition to Cochrane please add at least two more databases. (e.g., PubMed/MEDLINE and Embase).
---

	Materials and Methods section: Page 8/Lines 9-11: The protocol available in Open Science Framework was well documented and updated. Although it is not clear what methodology you followed to conduct an overview of reviews? Suggest authors refer to Cochrane Handbook Chapter V: Overviews of Reviews. https://training.cochrane.org/handbook/current/chapter-v#section--4 Also, if you followed PRISMA for reporting you should cite it in your methods section. http://www.prisma-statement.org/ Page 8/Lines 19-24: A comprehensive search of the literature should always include more than one database and the search strategy should be performed searching both keyword and subject indexing terms such as MeSH. For example, in addition to searching "surg*" in Title/Abstract the search should also include MeSH descriptor: [Surgical Procedures, Operative]. Although authors have described how they conducted the search in their methods section they should provide the complete search strategy in their supplemental documents. This ensures the reproducibility of the review. (see PRISMA for Searching requires you to "Include the search strategies for each database and information source, copied and pasted exactly as run" (see http://www.prisma-statement.org/Extensions/Searching) Page 8/Lines 29-45: Authors defined surgical and drug interventions, but they did not clearly state the inclusion criteria. Consider revising to clearly state your inclusion criteria similar to how they stated their exclusion criteria. This helps better understand how you screened the records. For example, you can state that "studies will be included if the met the following criteria..." Page 9/Lines 8-11: Please identify how the authors screened the records. Did you use a screening software (e.g., Covidence, Eppi-reviewer, DistillerSR). Did you export the records from Cochrane to a CSV (Excel) and conducted screening of title/abstract in Excel? Also can you report the inter-rater reliability showing the consistency of agreement between EAZ and JV? Page 10/Line 6: Please add reference to GRADE. https://www.gradeworkinggroup.org/ Page 10/Line 6-13: In addition to using GRADE, authors should include what tool they used to address risk of bias and perform critical appraisal of each review. For example, did they use AMSTAR? AMSTAR is a validated tool often used for overview of reviews/umbrella reviews? https://amstar.ca/Amstar_Checklist.php https://www.bmj.com/content/358/bmj.j4008 Please follow Cochrane V.4.9 Assessing methodological quality/risk of bias of included systematic reviews https://training.cochrane.org/handbook/current/chapter-v#section--4 Page 10/Line 19-43: Good inclusion of how they applied statistical models to help combine and standardized results from RCTS as well as the software used. Page 11/Line 3-24: Below are areas that the protocol as well as the manuscript needs clarification.
--	--

	How did the authors address missing information? Did they contact the corresponding author requesting clarification of missing/unclear data? Please include how authors addressed overlapping reviews which is a common issue with conducting overview of reviews that include the same primary studies. Especially if you are re-analyzing data it introduces risk of bias and double counting outcomes of a study. See Cochrane V.4.4 Managing overlapping systematic reviews https://training.cochrane.org/handbook/current/chapter-v#section--4-4 Results Section Page 12/Line 15-31: Authors should clarify what they mean by "manual inspection" of title/abstract. It is confusing why they needed to perform screening with just the word "surgery" in the title/abstract removing only 440 records. Did they then re-screen applying their i/e criteria to the remaining 2,055 records to help identify 223 eligible for full text review? It is unclear that independently, dual/blinded screening of 2,495 records was performed and this mixed method that included manual screening introduced bias in the selection process. Limitations section: Page 17/Lines 17-25: Authors should explain in more detail how they used the "algorithm" to filter papers. Was this performed in Excel or software tool? What was the algorithm used? Was it performed by one person? If it was it introduces bias in the selection process. Page 17/Lines 45-52: Although you highlighted as a limitation that you only used Cochrane Database there was no rationale to support restricting the search to only Cochrane. Suggest rerunning the search strategy in multiple databases to include both keywords and indexed terms (e.g., MeSH, Emtree) Figure 1/PRISMA Flow Diagram: If your Cochrane search resulted in 8,931 records and you only screened for 2,495 records how did the authors reduce the records by 6,436? It appears authors searched Cochrane reviews only and that would support screening of 2,495 records. The importance of including the full search strategy including limits/filters applied would permit you to show that you limited it to Cochrane reviews. This supports transparency and reproducibility. Example (surg*):ti,ab,kw (Word variations have been searched)" in Cochrane Reviews Missing reporting of duplicates: Your PRISMA flow diagram is missing a section requiring the reporting of how records were removed. How many records were duplicates? Were they removed for other reasons? The fact that you removed 6,436 records is a large number of records especially if most were not duplicates is problematic because it introduces potential bias in your process. Your process needs to be transparent to show how you minimized the bias in your selection of records. Records removed before screening: Duplicate records removed (n =) Records marked as ineligible by automation tools (n =) Records removed for other reasons (n =)
--	--

	(See http://www.prisma-statement.org/PRISMAStatement/FlowDiagram Page 47/PRISMA 2020 Checklist  - Line 5 Eligibility Criteria: Authors should be clear what their inclusion criteria is - Line 7 Search Strategy: Missing the full search strategies for database searches. Also need to search additional databases. - Line 8 Selection Process: Missing what methods used and tools/software for screening. - Line 10b Data Items: Need to address missing/unclear information. - Line 11 Study Risk of Bias Assessment: Although they indicated the two reviewers screened independently, they did not "specify the methods used to assess risk of bias" including what tools (e.g., software) was used. - Line 14 Reporting bias assessment: Need to add the risk of bias due to missing results in a synthesis. - Line 18 Risk of bias in studies: Did not present risk of bias for each included study. - Line 20a Results of synthesis: Did not briefly summarize the risk of bias among studies. - Line 21 Reporting biases: Need to add risk of bias due to missing results. Page 50/PRISMA 2020 Checklist  - Line 5 Risk of bias: Missing the risk of bias in the included articles
--	---

REVIEWER	Jin, James Middlemore Hospital, Department of Surgery
REVIEW RETURNED	17-Sep-2023

GENERAL COMMENTS	An interesting study, well done. As an umbrella review it highlights this area of evidence in a balanced and effective manner. The findings are presented clearly. The main limitation is that this study only includes Cochrane reviews, however expanding such a review to cover other databases would be highly unfeasible due to the volume of eligible articles encountered.
---

REVIEWER	Xiao, Mengli University of Colorado System
REVIEW RETURNED	16-Oct-2023

GENERAL COMMENTS	Comment about meta-analysis:  - The primary outcome is not clearly defined. - Methods: The exact methods the authors used to standardize effects are not clear. For example, did the author obtain the standardized effect size for continuous, binary, and survival outcomes? - Methods: What's the justification for a 99.5% confidence interval for the primary outcome? - Results: What's the distribution heterogeneity summary of included meta-analyses? - Results: What do the small study effect and publication bias look like among the included meta-analyses (Belbasis et al., 2022)? - Line 26, Page 11: should that be "Based on 99.5% confidence interval"? Reference: Belbasis, L., Bellou, V., & Ioannidis, J. P. (2022). Conducting umbrella reviews. BMJ medicine, 1(1).
---

VERSION 1 – AUTHOR RESPONSE

Reviewer: 1

Dr. Jitin Bajaj, NSCBM Medical College

Comments to the Author:

What was the main aim and hypothesis of the study?

We have rewritten and expanded the last paragraph of the Introduction, so as to be clear about the hypothesis and main aims of the study.

“A summary of the existing body, mapping the gaps of evidence on surgical versus medical interventions across diseases does not exist in the literature. A synthesis of this existing body of evidence is important to guide evidence-based care and inform decisions in the clinic where surgery and medical management are both reasonable options. We hypothesized that there may be a dearth of randomized evidence comparing surgery versus drugs and that even in topics where such RCTs exist the evidence provided by them may be weak. To find RCTs comparing surgical vs. pharmaceutical interventions, we conducted an umbrella review (an overview of systematic reviews) [9, 10] by searching the Cochrane Database of Systematic Reviews for reviews considering comparisons of surgery to drugs. We aimed to examine the prevalence of intended comparisons of surgery to drug regimens, how often such comparisons had any RCTs, and, whenever RCTs were available, what was the strength of evidence of such comparisons, and whether surgery or the drug intervention was favored.”

In the table 2 showing comparison between the medical and surgical arm for epilepsy surgery, why you have shown a low grade assessment even when three randomized controlled trials are already there showing surgery superior to the medical arm in drug-resistant epilepsy.

How did you defined the grade assessment?

The GRADE assessments are extracted from the included reviews which we have made more clear in the Methods section. We used the standard GRADE tool that is employed now routinely by Cochrane systematic reviews. Even in the presence of several RCT, the GRADE assessment can be “low”, in fact this is very common across Cochrane. We have double-checked the Cochrane review on epilepsy and indeed it scores this comparison as “low”.

Can you give references in the results of the respective RCTs which you found?

Regarding the availability of the raw data the respective RCTs are found in the repository of OSF under input and outcomes_RCT_basis.csv where we have the name of the study or first author year of publication as well as the journal identified, we have also further clarified the availability of the raw data further in the methods section. It would be outside the journal limits to add the references of all the RCTs in the main manuscript.

What benefit to the literature you seem to seek with your findings?

This study may possibly benefit the literature by showing that there is a gap in evidence that needs to be filled, considering that such a gap often renders evidence-based decision-making difficult. These issues are covered in the Discussion under the sections Conclusions (revised/expanded) and Future Research (manuscript p. 20).

Reviewer: 2

Ms. Gisela Butera, National Institutes of Health

Comments to the Author:

Overall comments: Overall, the review highlights the need to assess the level of evidence available on surgical versus medical treatments. The authors successfully described why they conducted an overview of reviews to identify the evidence but there are areas especially within the methodology and reporting of results that needs improvement.

Thank you very much for the kind appreciation of our work. We hope to have addressed your suggestions appropriately.

Introduction section:

Page 7: Authors nicely identified the gaps in the literature and the challenges in the quality of the evidence.

Thank you!

Page 7/Lines 44-51: There are concerns with searching only one database. If the aim of your review is to identify RCTs comparing surgical vs. pharmaceutical interventions, you should include additional databases. In addition to Cochrane please add at least two more databases. (e.g., PubMed/MEDLINE and Embase).

We have expanded and clarified that Cochrane was the most appropriate database to search given the nature of our umbrella review. Specifically, we state in the Limitations now that "only one database (Cochrane Database of Systematic Reviews) was used for this study, and we did not examine non-Cochrane meta-analyses published as journal articles. While the database aims to be all-inclusive, there are still some topics in medical and surgical care that have not been covered by Cochrane reviews. However, the Cochrane database is more meticulous in describing its methods and it will routinely publish systematic reviews that have found no eligible articles, while this is unlikely in systematic reviews published in traditional journals. Therefore, limiting our search to the Cochrane Database of Systematic Reviews has limited the problem of publication bias and also prevented a problem of overlapping systematic reviews. Furthermore, Cochrane is routinely used in clinical guideline development.

Materials and Methods section:

Page 8/Lines 9-11: The protocol available in Open Science Framework was well documented and updated. Although it is not clear what methodology you followed to conduct an overview of reviews?

Suggest authors refer to Cochrane Handbook Chapter V: Overviews of Reviews.

<https://training.cochrane.org/handbook/current/chapter-v#section--4>

We followed the methods of Belbasis et al. BMJ Medicine article for guidance for umbrella reviews and we have now specified this in the beginning of the Methods section. As suggested, we also refer now to the Cochrane Handbook Chapter V for more orientation of the reader.

Also, if you followed PRISMA for reporting you should cite it in your methods section.

<http://www.prisma-statement.org/>

Thank you for pointing that out we have added that to the first paragraph of the Methods section

Page 8/Lines 19-24: A comprehensive search of the literature should always include more than one database and the search strategy should be performed searching both keyword and subject indexing terms such as MeSH. For example, in addition to searching "surg*" in Title/Abstract the search should also include MeSH descriptor: [Surgical Procedures, Operative]. Although authors have described how they conducted the search in their methods section they should provide the complete search strategy in their supplemental documents. This ensures the reproducibility of the review. (see PRISMA for Searching requires you to "Include the search strategies for each database and information source, copied and pasted exactly as run" (see <http://www.prisma-statement.org/Extensions/Searching>)

Please see our reply above on why we believe that focusing the search on Cochrane reviews is justified and actually preferable in this topic, given our main aims. We have provided the specification of the search following your suggestions (surg*(ti;ab;kw)) in the introduction (first paragraph).

Page 8/Lines 29-45: Authors defined surgical and drug interventions, but they did not clearly state the inclusion criteria. Consider revising to clearly state your inclusion criteria similar to how they stated their exclusion criteria. This helps better understand how you screened the records. For example, you can state that “studies will be included if they met the following criteria...”

As we have clarified upfront “Inclusion criteria for reviews were consideration of RCTs and comparing a surgical to a drug intervention.” Then the exclusion criteria sharply demarcate which interventions are eligible.

Page 9/Lines 8-11: Please identify how the authors screened the records. Did you use a screening software (e.g., Covidence, Eppi-reviewer, DistillerSR). Did you export the records from Cochrane to a CSV (Excel) and conducted screening of title/abstract in Excel? Also can you report the inter-rater reliability showing the consistency of agreement between EAZ and JV?

We screened the records using Excel. Following your recommendation we assessed the inter-rater reliability of the independent screening showing a kappa of 0.36 and a 90% agreement on exclusion decisions at the first stage (before in-depth review). Considering our very broad inclusion process and the 3 category classification we consider this fair. All discrepancies were when classifying reviews in the category “unsure” and all these were resolved. We have included language about the inter-rater reliability both in the Methods section and in the Results section.

Page 10/Line 6: Please add reference to GRADE. <https://www.gradeworkinggroup.org/>
Thank you for pointing it out, we have added the reference.

Page 10/Line 6-13: In addition to using GRADE, authors should include what tool they used to address risk of bias and perform critical appraisal of each review. For example, did they use AMSTAR? AMSTAR is a validated tool often used for overview of reviews/umbrella reviews?

https://amstar.ca/Amstar_Checklist.php

<https://www.bmj.com/content/358/bmj.j4008>

We did not use AMSTAR as the primary aim of the study was not to look to make statements on the methodology of the systematic reviews themselves. Perhaps more importantly, Cochrane systematic reviews score high because AMSTAR-2 development was based explicitly on the Cochrane Handbook and we only include Cochrane reviews in our evaluation. We have added relevant language and citations on these issues in the Limitations section.

Please follow Cochrane V.4.9 Assessing methodological quality/risk of bias of included systematic reviews

<https://training.cochrane.org/handbook/current/chapter-v#section--4>

Please see response to comment above.

Page 10/Line 19-43: Good inclusion of how they applied statistical models to help combine and standardized results from RCTS as well as the software used.

Thank you.

Page 11/Line 3-24: Below are areas that the protocol as well as the manuscript needs clarification. How did the authors address missing information? Did they contact the corresponding author requesting clarification of missing/unclear data?

When extracting the raw data this was done directly from the completed systematic reviews in Cochrane, therefore the raw data was available for all reviews that had comparisons.

Please include how authors addressed overlapping reviews which is a common issue with conducting overview of reviews that include the same primary studies. Especially if you are re-analyzing data it introduces risk of bias and double counting outcomes of a study.

See Cochrane V.4.4 Managing overlapping systematic reviews

<https://training.cochrane.org/handbook/current/chapter-v#section--4-4>

We have now added in the Additions to the protocol that “We also examined whether the eligible RCTs that were included in the systematic reviews might have any overlap between different reviews.” In the Results section, we have added that there were 19 RCTs that appeared in more than one review of which 14 also had overlap in outcomes. Overlapping RCTs comprised >50% of the included RCTs in only 2 of 103 meta-analyses and even in these 2 meta-analyses there were additional non-overlapping RCTs.

Results Section

Page 12/Line 15-31: Authors should clarify what they mean by "manual inspection" of title/abstract. It is confusing why they needed to perform screening with just the word "surgery" in the title/abstract removing only 440 records. Did they then re-screen applying their i/e criteria to the remaining 2,055 records to help identify 223 eligible for full text review? It is unclear that independently, dual/blinded screening of 2,495 records was performed and this mixed method that included manual screening introduced bias in the selection process.

Page 17/Lines 17-25: Authors should explain in more detail how they used the "algorithm" to filter papers. Was this performed in Excel or software tool? What was the algorithm used? Was it performed by one person? If it was it introduces bias in the selection process.

To automatically filter the retrieved articles from the search we used a R script that is available on our github repository for going through the studies automatically to look for word variations of surg* in the title or abstract. This is because many of the studies were about pre-operative care or post-operative care as well as withdrawn. The other remaining articles were screened manually, independently using the title and abstract for inclusion and then in a last round with full text review to clear discrepancies. We have further clarified our study selection approach in our Methods section.

Page 17/Lines 45-52: Although you highlighted as a limitation that you only used Cochrane Database there was no rationale to support restricting the search to only Cochrane. Suggest rerunning the search strategy in multiple databases to include both keywords and indexed terms (e.g., MeSH, Emtree)

Thank you for this comment we have further clarified this in the Strengths and Limitations section and in the limitations section of the Discussion. As we state in the Limitations, “only one database (Cochrane Database of Systematic Reviews) was used for this study, and we did not examine non-Cochrane meta-analyses published as journal articles. While the database aims to be all-inclusive, there are still some topics in medical and surgical care that have not been covered by Cochrane reviews. However, the Cochrane database is more meticulous in describing its methods and it will routinely publish systematic reviews that have found no eligible articles, while this is unlikely in systematic reviews published in traditional journals. Therefore, including systematic reviews from journals may have distorted the picture and also caused a problem of overlapping systematic reviews.”

Figure 1/PRISMA Flow Diagram:

If your Cochrane search resulted in 8,931 records and you only screened for 2,495 records how did the authors reduce the records by 6,436? It appears authors searched Cochrane reviews only and that would support screening of 2,495 records. The importance of including the full search strategy

including limits/filters applied would permit you to show that you limited it to Cochrane reviews. This supports transparency and reproducibility.

Example

(surg*):ti,ab,kw (Word variations have been searched)" in Cochrane Reviews

The full Cochrane database included at the date of search 8931 records and the search retrieved 2495 articles. This should be clear in the revised version of the PRISMA flowchart.

Missing reporting of duplicates: Your PRISMA flow diagram is missing a section requiring the reporting of how records were removed. How many records were duplicates? Were they removed for other reasons? The fact that you removed 6,436 records is a large number of records especially if most were not duplicates is problematic because it introduces potential bias in your process. Your process needs to be transparent to show how you minimized the bias in your selection of records. We did not have any duplicates of Cochrane reviews.

Records removed before screening:

Duplicate records removed (n =) no duplicates

Records marked as ineligible by automation tools (n =) the 440 which we have specified in our PRISMA flowchart.

Records removed for other reasons (n =) none were removed for other reasons

(See <http://www.prisma-statement.org/PRISMAStatement/FlowDiagram>

Page 47/PRISMA 2020 Checklist

- Line 5 Eligibility Criteria: Authors should be clear what their inclusion criteria is See comment above
- Line 7 Search Strategy: Missing the full search strategies for database searches. Also need to search additional databases. See comment above
- Line 8 Selection Process: Missing what methods used and tools/software for screening. See comment above
- Line 10b Data Items: Need to address missing/unclear information. No missing information
- Line 11 Study Risk of Bias Assessment: Although they indicated the two reviewers screened independently, they did not "specify the methods used to assess risk of bias" including what tools (e.g., software) was used. We used the GRADE assessment from the included Cochrane review.
- Line 14 Reporting bias assessment: Need to add the risk of bias due to missing results in a synthesis. No missing results
- Line 18 Risk of bias in studies: Did not present risk of bias for each included study.
- Line 20a Results of synthesis: Did not briefly summarize the risk of bias among studies. We have extracted the risk of bias assessment for the RCTs from the included reviews and summarized this in the Results of the manuscript, as well as in the Methods section (last sentence).
- Line 21 Reporting biases: Need to add risk of bias due to missing results. No missing results.

Page 50/PRISMA 2020 Checklist

- Line 5 Risk of bias: Missing the risk of bias in the included articles: See the above comment

Reviewer: 3

Dr. James Jin, Middlemore Hospital

Comments to the Author:

An interesting study, well done. As an umbrella review it highlights this area of evidence in a balanced and effective manner. The findings are presented clearly. The main limitation is that this study only includes Cochrane reviews, however expanding such a review to cover other databases would be highly unfeasible due to the volume of eligible articles encountered.

Thank you for your kind appreciation of our work.

Reviewer: 4

Dr. Mengli Xiao, University of Colorado System

Comments to the Author:

Comment about meta-analysis:

- The primary outcome is not clearly defined.

We have tried to further specify the primary outcomes by revising the relevant section in the Methods.

- Methods: The exact methods the authors used to standardize effects are not clear. For example, did the author obtain the standardized effect size for continuous, binary, and survival outcomes?

We used the effects as provided by the Cochrane reviews. The standardization that we refer to pertains to the use of the same random effects method for synthesis employed in all meta-analyses.

- Methods: What's the justification for a 99.5% confidence interval for the primary outcome?

We follow the recommendation to move the threshold for typical statistical significance from $p=0.05$ to $p=0.005$ (referenced this in the methods section under statistical analysis Benjamin et al. 2018 Nat Hum Behavior).

- Results: What's the distribution heterogeneity summary of included meta-analyses?

The distribution of I-squared heterogeneity metric of the meta-analyses had a median of 43% (IQR = 0-80%) and we have added this information in the Results

- Results: What do the small study effect and publication bias look like among the included meta-analyses (Belbasis et al., 2022)?

As the included reviews' outcomes (~70%) did not contain more than 2 RCTs we could only perform Egger's for very few (the recommended number of articles being 10, which we only had for 5 outcomes of which 3/5 showed signs of significant publications bias). We performed test of excess significance as well on an outcome basis as well as for the entire field. The results have now been added to the Results section.

- Line 26, Page 11: should that be "Based on 99.5% confidence interval"?

No, as these are based on the 95% CI of the original reviews' meta-analysis

VERSION 2 – REVIEW

REVIEWER	Jin, James Middlemore Hospital, Department of Surgery
REVIEW RETURNED	10-Dec-2023
GENERAL COMMENTS	Despite its limitations the review is accurately and fairly conducted. The results are helpful in informing future research for randomised trials. In general there is a paucity of head to head randomised studies on surgical vs medical management of diseases. Further systematic reviews in this area should include other databases and focus on specific areas within surgery.